# Mitral Transcatheter Edge-to-Edge Repair and Clinical Value of Novel Echocardiographic Biomarkers: A Hypothesis-Generating Study

**DOI:** 10.3390/biomedicines12081710

**Published:** 2024-08-01

**Authors:** Javier Solsona-Caravaca, Rubén Fernández-Galera, Víctor González-Fernández, Lorenzo Airale, Johny Rivas, Luca Scudeler, Núria Vallejo, Gisela Teixidó-Turà, Guillem Casas, Filipa Valente, Ruper Oliveró, Yassin Belahnech, Gerard Martí, Bruno García, Ignacio Ferreira-González, José F. Rodríguez-Palomares, Laura Galian-Gay

**Affiliations:** 1Cardiology Department, Hospital Vall d’Hebron, 08035 Barcelona, Spain; javiersolsonacaravaca@gmail.com (J.S.-C.); victor.gonzalez@vallhebron.cat (V.G.-F.); guillem.casas@vallhebron.cat (G.C.); filipaxaviervalente@gmail.com (F.V.); ruperolivero26@gmail.com (R.O.); drgmarti@googlemail.com (G.M.); lauragaliangay@gmail.com (L.G.-G.); 2Internal Medicine Department, Città della Salute e della Scienza Hospital, 10126 Turin, Italy; lorenzo.airale@unito.it; 3Cardiology Department, Città della Salute e della Scienza Hospital, 10126 Turin, Italy; 4Centro de Investigación Biomédica en Red Enfermedades Cardiovasculares (CIBERCV), 28029 Madrid, Spain; 5Centro de Investigación Biomédica en Red de Epidemiología y Salud Pública (CIBERESP), 28029 Madrid, Spain

**Keywords:** mitral regurgitation, transcatheter edge-to-edge repair, heart failure hospitalization, cardiovascular death, myocardial work, ventricle–pulmonary artery coupling, left atrial strain

## Abstract

Background: Longitudinal data on reverse cardiac remodeling and outcomes after transcatheter edge-to-edge repair (TEER) are limited. Methods: A total of 78 patients with severe mitral regurgitation (MR) were included retrospectively. All patients had echocardiography at baseline and again six months after TEER. They were monitored for a primary composite endpoint, consisting of heart failure hospitalization and cardiovascular death, over 13 months. Results: Significant decreases in the left ventricular ejection fraction (LVEF), all myocardial work indices (except global wasted work), and the left atrial reservoir were observed after TEER. Additionally, there was a decrease in the pulmonary artery systolic pressure and an increase in the tricuspid annular plane systolic excursion/pulmonary artery systolic pressure (TAPSE/PASP) ratio. A post-TEER TAPSE/PASP ratio of <0.47 (HR: 4.76, *p*-value = 0.039), and a post-TEER left atrial reservoir of <9.0% (HR: 2.77, *p*-value = 0.047) were associated with the primary endpoint. Conclusions: Echocardiography post-TEER reflects impairment in ventricular performance due to preload reduction and right ventricle and pulmonary artery coupling improvement. Short-term echocardiography after TEER identifies high-risk patients who could benefit from a close clinical follow-up. The prognostic significance of LA strain and the TAPSE/PASP ratio should be validated in subsequent large-scale prospective studies.

## 1. Introduction

Mitral regurgitation (MR) is the second-most frequent valvular heart disease [1] and the underlying mechanism may be primary or secondary [1]. The main causes of primary MR are mitral valve prolapse and chordal rupture (flail leaflet) [2]. Secondary MR is seen in dilated or ischemic cardiomyopathies (ventricular MR) and patients with longstanding atrial fibrillation (atrial MR) [3].

Based on the European guidelines on the management of patients with valvular heart disease, transcatheter edge-to-edge repair (TEER) may be considered in symptomatic patients with severe primary MR at high operative risk (class IIb recommendation) [1]. On the other hand, TEER should be considered in symptomatic patients with severe secondary MR, despite optimal medical therapy, who are not appropriate for surgery (class IIa recommendation) [1,4,5,6].

Longitudinal data on reverse cardiac remodeling and outcomes after TEER are limited [7]. Given that the left ventricular ejection fraction (LVEF) might not be an optimal metric for left ventricular (LV) systolic performance in this population, novel echocardiographic biomarkers, such as left ventricular global longitudinal strain (LV GLS) and myocardial work (MW) indices, have been proposed [8]. Moreover, left atrial (LA) strain and right ventricle and pulmonary arterial (RV-PA) coupling have emerged as relevant prognostic indicators in patients with MR treated with TEER [7].

Importantly, heart failure readmissions and all-cause mortality during the first year after TEER remain substantial, occurring in nearly 17% of cases, even after a successful procedure [9,10].

Therefore, our study aimed to investigate the short-term impact of mitral TEER on LV performance and determine echocardiographic prognostic indicators to identify high-risk patients who could benefit from a close clinical follow-up.

## 2. Materials and Methods

### 2.1. Study Population

This was an observational retrospective longitudinal cohort study in patients with moderate-to-severe or severe MR treated with TEER in a tertiary hospital between 2020 and 2023. For this analysis, only patients with complete transthoracic echocardiography (TTE) at baseline (one month before TEER) and short-term follow-up (median: 6 months, interquartile range (IQR): 3 to 9 months) were included.

A total of 110 patients underwent TEER, 12 of whom did not have a comprehensive baseline TTE (Figure 1). Moreover, 20 patients had insufficient quality imaging or an incomplete TTE six months after TEER (Figure 1). As a result, we finally recruited 78 patients into the study: 49 (62.8%) patients with organic MR and 29 (37.2%) with functional MR (Figure 2).

Demographic and clinical information were collected from the patient’s electronic medical records. Patients were monitored for the primary composite endpoint, defined as heart failure hospitalization and cardiovascular death. The median follow-up period was 13 months after TEER (IQR: 5 to 20 months) (Figure 2).

We assessed the clinical value of novel echocardiographic indices obtained in baseline TTE and post-TEER TTE in identifying patients at high risk for heart failure hospitalization and cardiovascular death. Given the study’s small sample size and the limited number of events observed during the relatively short clinical follow-up period, our analysis should be considered as hypothesis-generating research.

This study was conducted in compliance with the “Good Clinical Practice” guidelines outlined in the Declaration of Helsinki and was approved by the Ethics Committee of the Vall d’Hebron Hospital (PR(AG)119/2024).

### 2.2. Echocardiography Data

Echocardiography exams were performed by expert imaging cardiologists, and measurements were obtained following the European Association of Cardiovascular Imaging (EACVI) standards [3,11,12] and confirmed by a senior expert. All TTE exams were performed with standard equipment (Vivid 95, General Electric Healthcare, Horten, Norway). Two expert and independent cardiologists blinded to the patients’ outcomes reviewed all images using EchoPAC Workstation.

The TTE data were obtained at baseline (one month before TEER) and after TEER (median 6 months, IQR: 3 to 9 months), including the MR grade, transmitral mean pressure gradient, LV diameters and volumes, LVEF, LV GLS, MW indices, LA volumes, LA strain, tricuspid regurgitation (TR) grade, tricuspid annular plane systolic excursion (TAPSE), right ventricular strain free wall (RV FWS), pulmonary artery systolic pressure (PASP), and the TAPSE/PASP ratio.

According to the EACVI recommendations, MR was classified as grade 1, 2, 3, or 4 [3]. Procedural success was defined as reducing the severity of the MR to grade 2 or less in TTE after TEER [5]. TEER complications were classified as either procedure-related or device-related [13].

The LV diameters were measured on the parasternal axis, and the LV volumes were measured in apical four- and two-chamber views. The LVEF was calculated using Simpson’s biplane method, while the LA volume was calculated using the biplane method. Speckle tracking echocardiography was used to assess the LV and LA strains. The LV GLS was obtained using apical four-chamber, two-chamber, and apical long-axis views [11]. The atrial strain was calculated in the reservoir phase using four- and two-chamber views [12].

The assessment of the LV MW incorporates blood pressure measurements to estimate the LV afterload [4]. MW is measured from pressure–strain loop areas that are constructed from the LV pressure curves combined with the LV GLS [14]. The peak systolic LV pressure is assumed to be equal to the peak arterial pressure recorded from the brachial cuff systolic pressure before the echocardiographic study in a supine position.

MW comprises multiple components: the global work index (GWI), global constructed work (GCW), global wasted work (GWW), and global work efficiency (GWE) [15,16]. The NORRE study provides useful reference ranges for novel indices of noninvasive MW in healthy volunteers over a wide range of ages and genders [17]. We can see an example of MW evaluation in Figure 3. 

According to the EACVI recommendations, TR is classified as grade 1, 2, 3, or 4 [3]. Right ventricular function was calculated using the TAPSE and RV FWS. The right atrial–right ventricular systolic gradient was determined from the peak velocity of the TR jet using the simplified Bernoulli equation [9]. The right atrial pressure was estimated by the diameter and collapsibility of the inferior vena cava and added to the calculated gradient to yield the PASP [9]. The TAPSE/PASP was then obtained [9], being a noninvasive index to assess the RV-PA coupling [18].

### 2.3. Statistical Analysis

The distribution of the variables was analyzed through graphical evaluation (histogram and Q–Q graph) and the Shapiro–Wilk test. Continuous variables were expressed as the means and standard deviations when the normality assumption was met, whereas those not following a normal distribution were presented as the medians and IQR ranges. Continuous variables were compared using Student’s *t*-test or the Mann–Whitney U-test, as appropriate. Categorical variables were expressed as absolute numbers and percentages and were compared using the chi-square test or Fisher’s exact test, as appropriate.

To analyze the differences between six months post-TEER and baseline pre-TEER imaging markers, we applied a paired *t*-test, a Wilcoxon test, or a McNemar test, as appropriate.

Cox regression analysis assessed the association between the clinical and echocardiographic parameters with the composite endpoint of “heart failure hospitalization and cardiovascular death.” Youden’s index was used to determine the cut-off values for the echocardiography markers that best predict the composite endpoint. Survival curves for the post-TEER TAPSE/PASP ratio and post-TEER LA reservoir strain were constructed using the Kaplan–Meier method; survival was assessed starting from after the TEER echocardiographic reassessment.

A *p*-value < 0.05 indicates statistical significance. Statistical analysis was performed using R Core Team (version 4.3.2 for Mac OSX).

## 3. Results

### 3.1. Baseline Characteristics of the Cohort

The mean age was 74 ± 9 years, and 46% were females. Most patients were symptomatic; 65.4% were New York Heart Association (NYHA) class III to IV. All patients were on optimal medical treatment before TEER (Table 1).

The median effective regurgitant orifice area (EROA) was 37.0 mm^2^, IQR: 30.0–40.0 mm^2^. Primary or organic MR was the main etiology (49 patients, 62.8%), while functional or secondary etiology was less frequent (29 patients, 37.2%). Within the functional MR subgroup, we included patients with atrial MR (11 patients, 37.9%) and ventricular MR (18 patients, 62.1%). The clinical and echocardiographic baseline characteristics are shown in Table 1 and Table 2.

A total of 49 patients (62.8%) had known atrial fibrillation, and 54 (69.2%) had prior heart failure hospitalization. No differences were observed in the heart failure hospitalization rate between the patients with organic and functional MR (71.4% versus 65.5%, *p*-value = 0.826). A total of 33 patients (42.3%) had chronic kidney disease, and 12 (15.3%) had chronic obstructive pulmonary disease.

Before the percutaneous procedure, 20 patients (25.6%) had an MR grade of III, while 58 patients (74.4%) had grade IV MR. The patients with organic MR exhibited more severe MR than the patients with functional MR (mean EROA of 40 mm^2^ versus 30 mm^2^, *p*-value = 0.038).

The median LVEF was 50.0% (IQR: 36–60%). In patients with organic MR, the LV volumes were larger than those with functional MR, particularly the LV end-diastolic volume (LVEDV) and indexed LV end-diastolic volume (LVEDVI); Table 2. No differences were detected in the MW indices (GWI, GCW, GWW, and GWE) at baseline between the patients with organic and functional MR (Table 2). The LV GLS (−13.76 ± 3.88%) and LA strain (mean of 10%, IQR: 7.00–16.0%) were reduced at baseline.

In the functional MR cohort, the patients with atrial MR had a higher LVEF and lower volumes than the patients with ventricular MR: mean LVEF of 58.0% (IQR: 55.0–62.0%) versus 40.5% (IQR: 34.2–54.0%), *p*-value = 0.003; mean LVEDVI of 89 mL (IQR: 79–110 mL) versus 115 mL (IQR: 100–184 mL), *p*-value = 0.038.

There were no significant differences in the baseline TAPSE, PASP, or TAPSE/PASP ratio values between the patients with organic and functional MR (Table 2). Of the patients treated with mitral TEER, 65.3% did not present significant TR (≤2) before the procedure.

### 3.2. Safety and Effectiveness of Mitral TEER in Our Population

Different devices were used for the TEER procedure, primarily MitraClip in 96% of cases and Pascal in 4%. Thirty percent of the cohort required more than one device to effectively treat MR.

A significant reduction in the MR grade was observed in the overall population after TEER (*p*-value < 0.001). TEER was successful in 51 patients (65.4%), without significant differences between the patients with organic and functional MR regarding the procedure’s rate success (65.3% versus 65.5% *p*-value = 0.691). Following TEER, the transmitral mean gradient consistently increased (2.21 ± 0.83 mmHg before TEER versus 3.95 ± 1.94 after TEER, *p*-value < 0.001).

In the present study, eight complications were reported (10.2%). Three of these complications (3.8%) were procedure-related, with two cases of tamponade and one vascular access site complication. Five (6.4%) were device-related: two patients had a chordal rupture and required urgent surgery, two other patients experienced partial clip detachment and needed a second percutaneous procedure, and there was one case of leaflet perforation treated conservatively due to high surgical risk.

### 3.3. Short-Term Impact of TEER on Left Ventricular Remodeling

Table 3 and Figure 4 show changes in the LV, LA, and right ventricular performance six months after TEER (IQR: 3 to 9 months). A significant reduction in the LVEF was observed (delta –3.63 ± 8.18, *p*-value < 0.001). Decreased LVEF after correction of MR occurs because of a greater decline in the LV end-diastolic diameter (LVEDD) and LVEDV compared to the LV end-systolic diameter (LVESD) and LV end-systolic volume (LVESV), Table 3.

Moreover, an impairment in all myocardial indices (GWI, GCW, and GWE) was observed following the acute reduction in the LV preload, except for GWW (Table 3). The LV GLS also tended to decrease after TEER, although it did not reach statistical significance (−13.28 ± 3.82 versus −12.28 ± 3.97; *p*-value = 0.085). The LA reservoir strain also diminished after TEER (12.03 ± 6.38 versus 10.00 ± 4.56; *p*-value = 0.002).

No changes in the right ventricular function (TAPSE, RV FWS) were reported after TEER (Table 3). A significant and early decrease in the PASP values after the percutaneous procedure was observed (53.64 ± 18.64 mmHg before TEER versus 43.86 ± 14.92 mmHg after TEER; *p*-value < 0.001).

Furthermore, an increase in the TAPSE/PASP was observed (0.39 ± 0.19 before TEER versus 0.45 ± 0.18 after TEER; *p*-value = 0.021), translating as beneficial changes in RV-PA coupling.

### 3.4. Impact of TEER at Follow-Up: Heart Failure Hospitalizations and Cardiovascular Death after TEER

The primary composite endpoint occurred in 27 patients (34.6%). There were no differences in the primary endpoint between the patients with organic and functional MR: twenty events (40.8%) versus seven events (24.1%), *p*-value = 0.211 (Figure 5). 

The association between clinical and echocardiographic variables with the primary endpoint (univariable analysis) is shown in Table 4. A higher risk of HF hospitalization or cardiovascular death was present among the patients with diabetes (HR: 2.25, 95% CI: 1.02–4.98, *p*-value = 0.045) and those with prior heart failure hospitalization (HR: 3.69, 95% CI: 1.26–10.83, *p*-value = 0.017).

In addition, a lower reservoir value of the post-TEER LA strain (*p*-value = 0.024) and a lower post-TEER TAPSE/PASP ratio (*p*-value = 0.038) were also associated with the primary endpoint, which was not verified when the TAPSE and PASP were analyzed separately (Table 4).

In the Kaplan–Meier survival analysis (Figure 6), patients with a TAPSE/PASP ratio of <0.47 had a higher risk of reaching the primary composite endpoint (HR: 4.76, 95% CI: 1.08–21.02, *p*-value = 0.039). Moreover, patients with a post-TEER LA reservoir of <9.0% also had an increased risk of reaching the primary composite endpoint (HR: 2.77, 95% CI: 1.01–7.59, *p*-value = 0.047).

When MR etiology was introduced into the Cox multivariate regression analysis, a post-TEER TAPSE/PASP ratio of <0.47 remained an independent prognostic indicator of the primary composite endpoint (HR: 4.67, 95% CI: 1.04–20.93, *p*-value = 0.043), together with a post-TEER LA reservoir of <9.0% (HR: 3.08, 95% CI: 1.08–8.76, *p*-value = 0.034).

## 4. Discussion

TEER is a safe and effective treatment in symptomatic patients with MR [1]. However, data on reverse cardiac remodeling and outcomes after TEER are limited. We evaluated the short-term impact of mitral TEER on cardiac performance using novel echocardiographic biomarkers, such as LV strain, MW, the TAPSE/PASP ratio, or LA strain. An impairment in ventricular performance due to preload reduction and RV-PA coupling improvement was observed at short-term follow-up after TEER. Additionally, we evaluated the clinical value of these novel echocardiographic indices for risk stratification and treatment guidance. The TAPSE/PASP ratio and LA reservoir appeared to be relevant independent prognostic indicators in patients with MR who were treated with TEER.

Contrasting with previously published studies, the present study mainly included patients with organic MR (62.8%), with functional MR being less frequent (37.2%). The EVEREST trial included patients with organic and functional MR [20]. However, organic MR only represented 41% of the patients [20]. The COAPT trial only included patients with functional MR, and patients with predominant atrial MR without LV dysfunction were excluded [21]. So far, there is little evidence in the literature regarding clinical outcomes of TEER in series with a predominance of organic MR patients.

In the present cohort, patients with organic MR exhibited higher ventricular volumes (LVEDV and LVEDVI) than those with functional MR. Moreover, slightly lower LV volumes were observed compared to the EVEREST trial patients, and significantly lower LV volumes and higher LVEF were observed compared to the COAPT trial patients (Appendix A in the Appendix A).

Distinct reasons can explain these differences. Firstly, the present functional MR cohort had a significant prevalence of patients without cardiomyopathy (37.9%) due to an atrial mechanism of the MR. Importantly, these patients are characterized by preserved LV function and LV volumes, in contrast to patients with ventricular MR [22]. Furthermore, ventricular functional MR was only treated in patients with “disproportionate” MR, characterized by severe MR, an acceptable LVEF, and non-markedly dilatated ventricles.

TEER was successful in 51 patients (65.4%) with a residual MR grade 1 or 2 at short-term follow-up. Notably, no significant differences regarding the procedure’s success were observed between the patients with organic and functional MR. These results are similar to the Spanish real-life registry, which showed a maintained MR reduction (grade 1 or 2) in 73% of patients 12 months after TEER [10].

The primary composite endpoint (heart failure hospitalization and cardiovascular death) occurred in 27 patients (34.6%). As in the previously mentioned registry [10], there was no difference in the primary endpoint between patients with organic or functional MR.

**LVEF, myocardial work indices, and LV remodeling**.

As expected, a significant decrease in the LVEF was observed after TEER. After MR correction, patients with preserved baseline LVEF may experience a decrease in the LVEF due to a greater reduction in the LVEDV compared to the LVESV [23,24].

Regarding myocardial indices, different studies have evaluated changes after TEER [5,14,25,26]. All myocardial indices (GWI, GCW, and GWE) were impaired in the short-term evolution in the present series, except for the GWW, due to preload reduction. Similar results were published by Galli et al. for patients with organic MR [25]. Yedidya et al. noted that the forward stroke volume index was the only echocardiographic parameter that improved six months after TEER in patients with functional MR [26].

At an extended follow-up, Galli et al. reported that the GLS, GWI, and GWE recovered one year after TEER, but not the GWW [25]. The fact that the GWW remained impaired might be attributable to the persistence of myocardial fibrosis [27,28]. GWW could be higher in patients with advanced disease, and it could delay reverse LV remodeling, negatively impacting the LV response to MR correction [25], with worse long-term survival [4]. Papadopoulos et al. also reported long-term beneficial changes in ventricular remodeling after TEER, with a reduction in the LV volumes and significant improvement of the GWI and GCW in patients with functional MR [14]. The GWI and GCW were also independently associated with worse long-term survival [4].


**RV-PA coupling: changes after TEER and clinical outcomes.**


In our series, we reported a significant and early decrease in the PASP value and an increase in the TAPSE/PASP ratio after TEER, similar to Adamo M. et al., who reported that 66% of patients enhanced their TAPSE/PASP ratio at the short-term follow-up after TEER [18]. This fact has important clinical significance because lower pulmonary pressure is one of the main determinants of reverse LV remodeling, as described by Cimino et al. [6]. Additionally, an improvement in the TAPSE/PASP ratio after TEER was independently associated with a reduced risk of mortality at the long-term follow-up [18].

In our series, patients with a post-TEER TAPSE/PASP ratio of <0.47 had a higher risk of reaching the primary composite endpoint. An impaired TAPSE/PASP ratio after TEER could be attributed to an increase in PASP, which may be partially explained by the persistence of significant residual MR after TEER due to a non-optimal procedure [18], higher postprocedural mitral mean gradients [18], or associated LV systolic or diastolic dysfunction. In addition, a reduced TAPSE/PASP ratio after TEER could be explained by a decrease in TAPSE in patients who are treated in an advanced stage of the disease accompanied by right ventricle systolic dysfunction. The prognostic significance of the TAPSE/PASP ratio needs to be validated in subsequent large-scale prospective studies.


**LA strain in patients treated with TEER and clinical outcomes.**


Increasing the preload can enhance reservoir function in the initial stages of MR when the LA is not rigid [23]. However, the LA strain decreases with increasing MR grades [23,29]. In the present cohort, a reduced baseline LA strain was observed, suggesting that the patients had advanced disease. In cases of LA stiffness, the left atrium cannot compensate for the change in preload, leading to hemodynamic failure, such as pulmonary edema [23,30].

Previous studies reported that the TAPSE/PASP ratio presented a significant negative correlation with the baseline LA v-wave [9]. Moreover, an improvement in RV-PA coupling was linked to an incremental LA reservoir [9]. However, RV-PA coupling progress was not associated with beneficial changes in LA function in our study. In fact, the LA reservoir decreased significantly after TEER and was accompanied by a reduction in the LV GLS, suggesting that the LA reservoir mainly depends on the LV longitudinal function [23]. In addition, LA strain impairment after TEER may result from a residual MR grade 2 or higher, which makes the LA reservoir decline further, as observed by Gucuk Ipek et al. [23].

In our series, patients with a post-TEER LA reservoir of <9.0% had a higher risk of reaching the primary composite endpoint. A deteriorated LA strain may indicate advanced and irreversible LA dysfunction, negatively affecting the prognosis [9,23,30]. The prognostic significance of LA strain should be validated in subsequent large-scale prospective studies.


**Risk stratification with echocardiographic markers.**


Due to their prognostic implications, patients with a deteriorated TAPSE/PASP ratio or LA reservoir in short-term TTE after TEER could benefit from close clinical follow-up. In this context, short-term echocardiography performed by imaging experts after TEER is essential to improve risk stratification and identify high-risk patients who could benefit from stricter monitoring and more aggressive medical treatment during clinical follow-up.

On the other hand, early intervention for MR seems crucial to prevent adverse LV remodeling. It would be interesting to identify TTE patients at baseline who could benefit from an anticipated TEER and avoid futility in patients already with advanced-stage disease. However, our study was not able to determine echocardiographic prognostic predictors in the baseline TTE, probably due to a low statistical power. In contrast, Trejo-Velasco et al. found that patients with MR treated with TEER had higher rates of heart failure readmissions and all-cause mortality if the baseline TAPSE/PASP ratio was ≤0.35 [9]. Additionally, Stassen et al. published that a more preserved baseline LA reservoir (≥9.8%) in patients with functional MR was independently associated with lower all-cause mortality [29].


**Study limitations.**


This retrospective, single-center, real-life study comprised a limited and heterogeneous population with MR caused by different etiologies and a relatively short follow-up period. In future studies, we will focus on single-etiology MR as the patients with organic MR, ventricular functional MR, and atrial functional MR exhibited differentiated physiopathology and likely a distinct clinical evolution. The retrospective study design may include a selection bias because some patients were excluded due to the insufficient quality of their echocardiography imaging.

The ventricular volumes were calculated by Simpson’s biplane method and not by 3D echocardiography. MW combines longitudinal myocardial deformation and afterload measures [31]. Despite this advantage, pressure–strain analysis does not differentiate between forward and regurgitant volume, which remains a limitation for assessing LV function in MR [31]. For this reason, the forward stroke volume index should be considered in future studies [26]. Moreover, there is only one vendor platform with a noninvasive MW algorithm (GE Healthcare) [15,16]. This fact and the great variety of MW parameters might represent a limitation for the broad application of MW in clinical practice [15,16].

Because of the study’s small sample size and the limited number of events observed during the short clinical follow-up period, our analysis serves as hypothesis-generating research. Our results may encourage further studies in larger cohorts with extended follow-up periods.

## 5. Conclusions

Post-TEER echocardiography reflects impairment in ventricular performance (LVEF, LV GLS, and most MW indices) due to preload reduction. However, there was a decrease in PASP and an increase in the TAPSE/PASP ratio, translating into beneficial changes in RV-PA coupling.

We hypothesize that the TAPSE/PASP ratio and the LA reservoir obtained in short-term TTE after TEER are important prognostic indicators that could be associated with a higher rate of adverse events (heart failure hospitalization and cardiovascular death). Therefore, post-TEER echocardiography is crucial to identify high-risk patients who could benefit from close clinical follow-up.

Because of the study’s small sample size and the limited number of events observed during the short clinical follow-up period, the prognostic significance of LA strain and the TAPSE/PASP ratio should be validated in subsequent large-scale prospective studies.

## Figures and Tables

**Figure 1 biomedicines-12-01710-f001:**
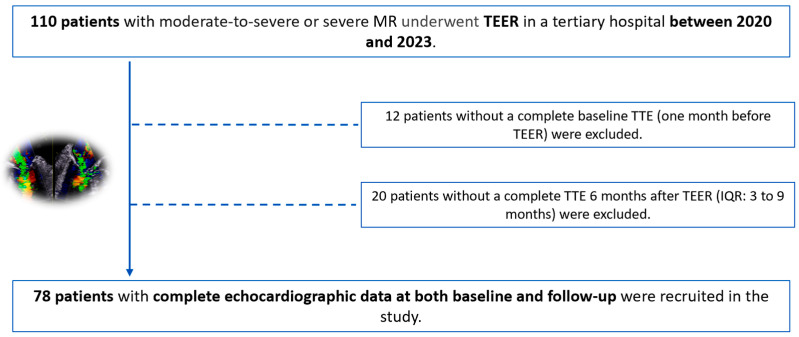
Study flowchart. Thirty-two patients were excluded due to an incomplete echocardiographic evaluation or insufficient quality echocardiography imaging.

**Figure 2 biomedicines-12-01710-f002:**
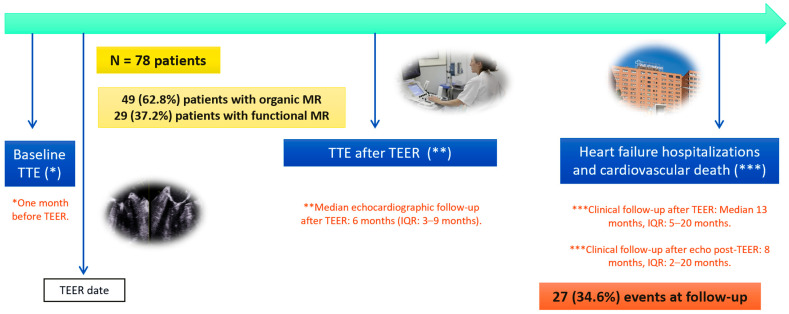
Patient follow-up after TEER. The median echocardiographic evaluation follow-up period was six months after TEER. The median clinical follow-up period was 13 months after TEER.

**Figure 3 biomedicines-12-01710-f003:**
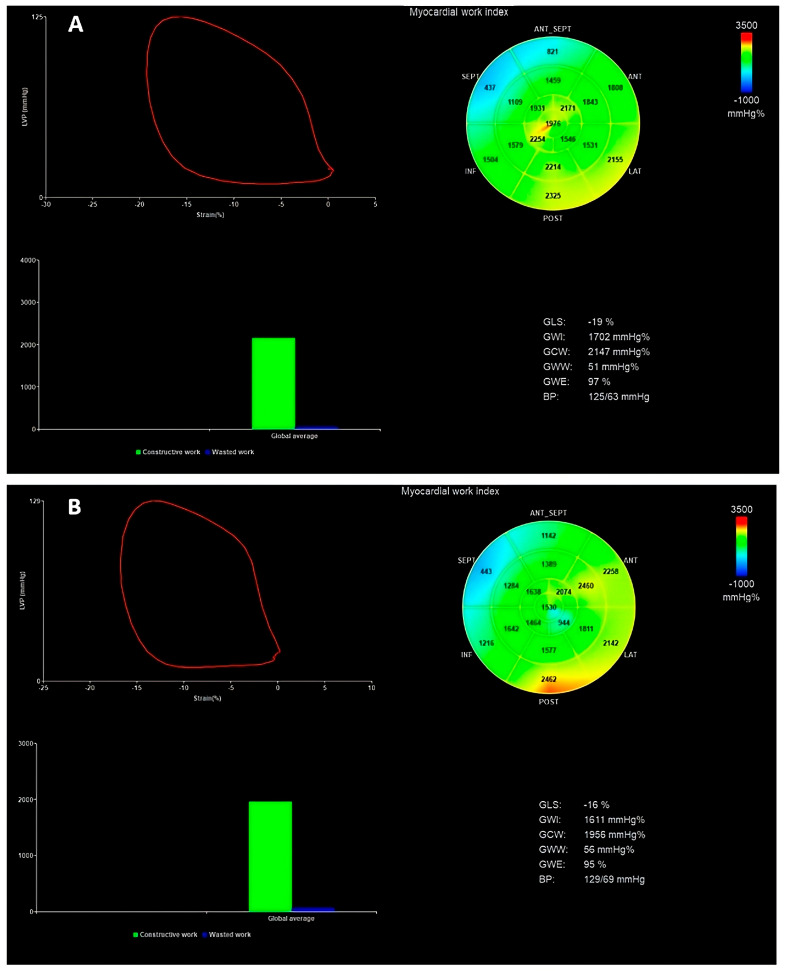
(**A**) In the first panel, the MW indices were evaluated before TEER. (**B**) In the second panel, the MW indices were evaluated six months after TEER. In this case, an impairment of strain and all myocardial indices can be seen (except GWW) after TEER due to an acute reduction in the LV preload. GWI is defined as the total work within the area of the LV pressure–strain loop (area circled in red); GCW is the MW during segmental shortening in the systole and segmental lengthening during the isovolumic relaxation time; GWW is the work performed during lengthening in the systole and shortening during isovolumic relaxation; GWE is the ratio between the GCW divided by the sum of the GCW and GWW [15,16]. MW, myocardial work; GLS, global longitudinal strain; GWI, global work index; GCW, global constructive work; GWW, global wasted work; GWE, global work efficiency; BP, blood pressure.

**Figure 4 biomedicines-12-01710-f004:**
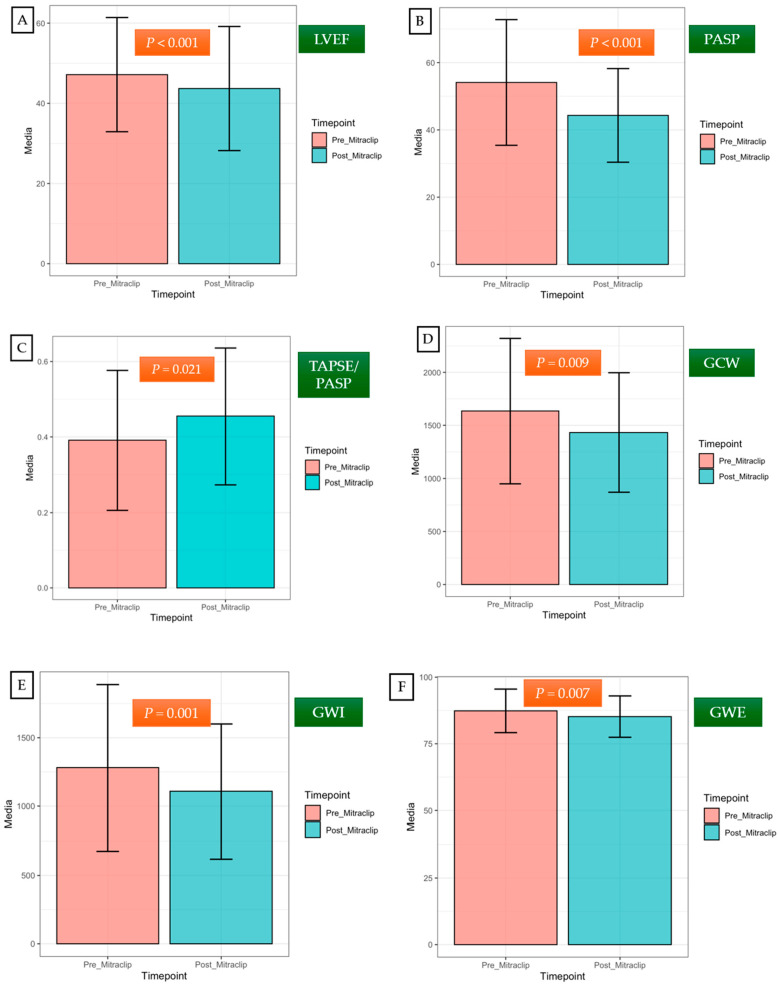
Short-term changes in left and right ventricular performance after TEER (media 6 months, IQR: 3 to 9 months). After TEER, a significant decrease in LVEF was observed (**A**). Significant and early improvements in the PASP values and the TAPSE/PASP ratio were noted (**B**,**C**). Impairments in GWI, GCW, and GWE were detected following the acute reduction in the LV preload (**D**–**F**). Error bars on the graphs represent the standard deviation of a dataset relative to the mean. LVEF, left ventricular ejection fraction; GWI, global work index; GCW, global constructive work; GWE, global work efficiency; TAPSE, tricuspid annular plane systolic excursion; PASP, pulmonary artery systolic pressure.

**Figure 5 biomedicines-12-01710-f005:**
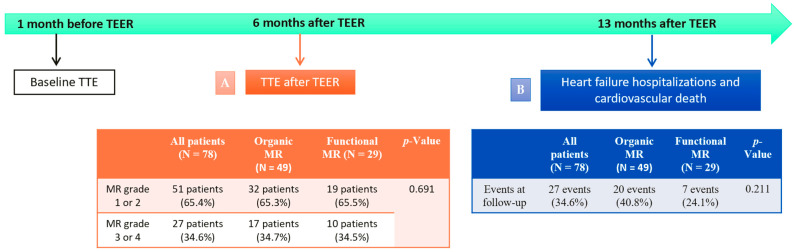
(**A**) Effectiveness of mitral TEER in patients with organic and functional MR (TTE performed a median of 6 months after TEER). (**B**) Heart failure hospitalizations and cardiovascular death (after a median follow-up of 13 months) in patients with organic and functional MR.

**Figure 6 biomedicines-12-01710-f006:**
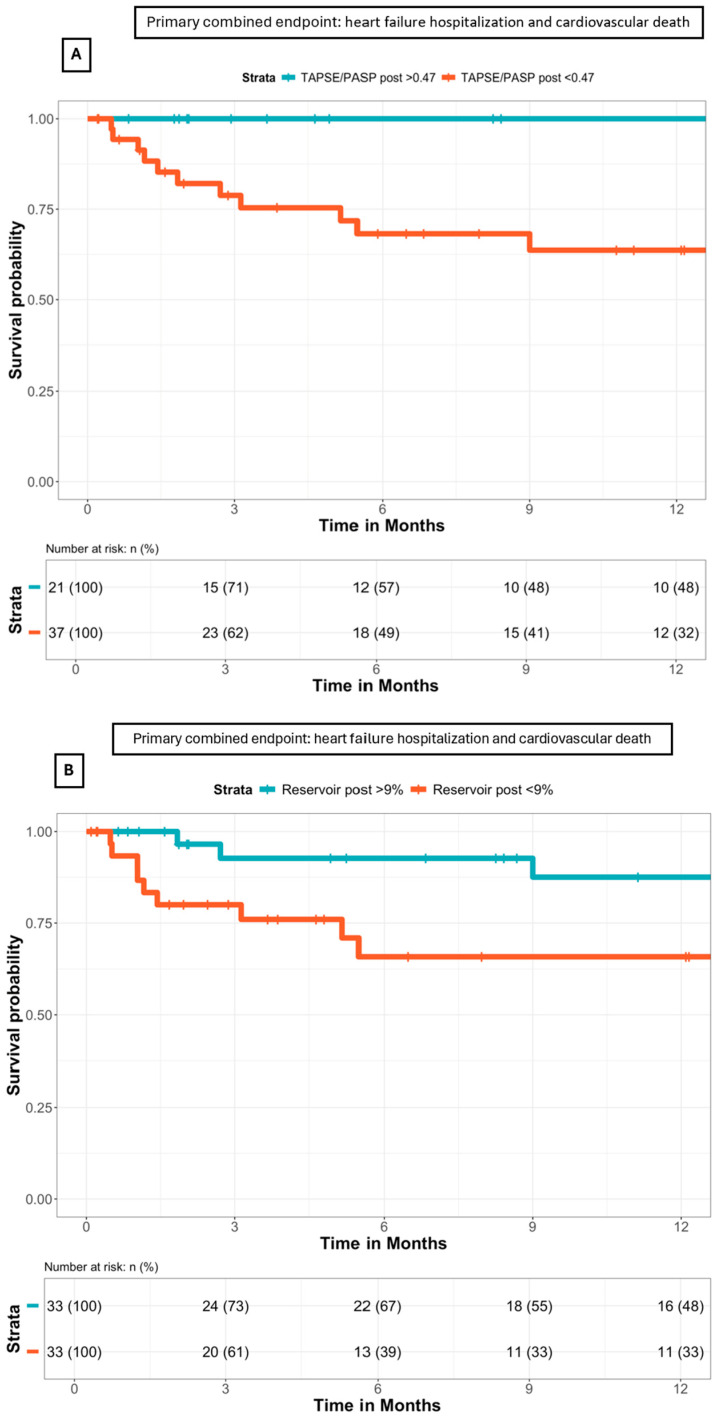
(**A**) Survival analysis according to the post-TEER TAPSE/PASP ratio, as shown by Kaplan–Meier curves. A post-TEER TAPSE/PASP ratio of <0.47 was associated with higher rates of the primary composite endpoint (*p*-value = 0.039). (**B**) Survival analysis according to the post-TEER LA reservoir, as shown by Kaplan–Meier curves. An LA reservoir of <9.0% was associated with higher rates of the primary composite endpoint (p-value = 0.047).

**Table 1 biomedicines-12-01710-t001:** Clinical baseline characteristics of the cohort (one month before TEER). NYHA, New York Heart Association; ACE-I, angiotensin-converting enzyme inhibitors; ARA, angiotensin receptor antagonist; MRA, mineralocorticoid receptor antagonists; SGLT2, sodium-glucose co-transporter 2.

	All Patients (N = 78)	Functional MR (N = 29)	Organic MR (N = 49)	*p*-Value
Age (year)	74 ± 9	75 ± 9	74 ± 9	0.898
Sex female	36 (46.2%)	16 (55.2%)	20 (40.8%)	0.320
Dyslipidemia	46 (58.9%)	19 (65.5%)	33 (67.3%)	1.000
Diabetes	34 (43.5%)	10 (34.4%)	23 (46.9%)	0.440
Hypertension	66 (84.6%)	25 (86.2%)	41 (83.6%)	1.000
Known atrial fibrillation	49 (62.8%)	24 (82.7%)	27 (55.1%)	0.052
Ischemic heart disease	36 (46.1%)	9 (31.0%)	26 (53.0%)	0.124
Prior heart failure hospitalization	54 (69.2%)	19 (65.5%)	35 (71.4%)	0.826
Pacemaker or defibrillation therapy	6 (7.6%)	2 (6.8%)	4 (8.1%)	1.000
Prior cardiac surgery	18 (26.5%)	7 (30.4%)	11 (24.4%)	0.811
NYHA class:				0.353
NYHA 1	1 (1.3%)	0 (0.0%)	1 (2.0%)	
NYHA 2	26 (33.3%)	13 (44.8%)	13 (26.6%)	
NYHA 3	43 (55.1%)	14 (48.3%)	29 (59.2%)	
NYHA 4	8 (10.3%)	2 (6.90%)	6 (12.2%)	
Chronic renal impairment	33 (42.3%)	10 (34.4%)	22 (44.8%)	0.644
Chronic obstructive pulmonary disease	12 (15.3%)	5 (17.2%)	7 (14.2%)	0.738
Furosemide	71 (91.0%)	26 (89.6%)	44 (89.7%)	1.000
Beta-blockers	58 (74.3%)	24 (82.7%)	35 (71.4%)	0.583
ACE-I, ARA-2, or sacubitril/valsartan	38 (48.7%)	15 (51.7%)	23 (46.9%)	0.866
MRA	33 (42.3%)	10 (34.4%)	23 (46.9%)	0.451
SGLT2-inhibitors	19 (24.3%)	11 (37.9%)	9 (18.3%)	0.137

**Table 2 biomedicines-12-01710-t002:** Echocardiographic baseline characteristics of the cohort (one month before TEER). Significant *p*-values are in bold. LVEF, left ventricular ejection fraction; LVED, left ventricular end-diastolic volume; LVEDVI, indexed left ventricular end-diastolic volume; LVESV, left ventricular end-systolic volume; LVEDD, left ventricular end-diastolic diameter; LVESD, left ventricular end-systolic diameter; LV GLS, left ventricular global longitudinal strain; GWI, global work index; GCW, global constructive work; GWW, global wasted work; GWE, global work efficiency; MR, mitral regurgitation; EROA, effective regurgitant orifice area; TAPSE, tricuspid annular plane systolic excursion; RV FWS, right ventricular free wall longitudinal strain; TR, tricuspid regurgitation; PASP, pulmonary artery systolic pressure.

	All Patients (N = 78)	Functional MR (N = 29)	Organic MR (N = 49)	*p*-Value
LVEF (%)	50.0 [36.0; 60.0]	54.0 [40.0; 60.0]	45.0 [35.0; 61.0]	0.427
LVEDV (mL)	120 [90.8; 151]	102 [84.0; 129]	127 [96.0; 153]	**0.048**
LVEDVI (mL/m^2^)	74.5 ± 29.1	60.1 [45.1; 72.9]	74.1 [62.6; 84.6]	**0.016**
LVESV (mL)	61.0 [38.0; 91.0]	48.0 [34.2; 72.8]	66.0 [47.0; 96.5]	0.065
LVEDD (mm)	54.9 ± 8.87	54.0 ± 7.46	55.4 ± 9.54	0.484
LVESD (mm)	38.0 [31.0; 49.0]	36.0 [32.0;44.0]	40.5 [31.0; 49.8]	0.361
LV GLS (%)	−13.76 ± 3.88	−13.68 ± 3.28	−13.80 ± 4.17	0.914
GWI (mmHg%)	1277 ± 600	1281 ± 476	1275 ± 654	0.966
GCW (mmHg%)	1628 ± 680	1615 ± 489	1635 ± 756	0.898
GWW (mmHg%)	149 [96.5; 218]	119 [59.0; 184]	157 [105; 225]	0.094
GWE (%)	90.0 [84.0; 93.5]	90.0 [87.0; 95.0]	89.0 [81.8; 93.0]	0.124
Left atrial volume index (mL/m^2^)	40.6 ± 19.3	40.4 [30.8; 45.8]	34.7 [26.4; 46.8]	0.675
Left atrial strain reservoir (%)	10.0 [7.00; 16.0]	8.00 [7.00; 11.0]	11.0 [7.25; 16.8]	0.118
MR grade:				0.691
MR grade 3	20 (25.6%)	10 (34.5%)	9 (18.4%)	
MR grade 4	58 (74.4%)	19 (65.5%)	40 (81.6%)	
EROA (mm^2^)	37.0 [30.0; 40.0]	30.0 [29.2; 40.0]	40.0 [30.0; 40.0]	**0.038**
Transmitral mean pressure gradient (mmHg)	2.05 [1.60; 2.68]	1.90 [1.35; 2.50]	2.20 [1.65; 2.75]	0.269
TAPSE (mm)	17.9 ± 3.79	16.9 ± 3.02	18.3 ± 4.06	0.113
RV FWS (%)	−19.20 [−21.95; −14.20]	−19.10 [−20.25; −14.80]	−19.40 [−25.00; −14.50]	0.381
Tricuspid regurgitation grade:				0.083
TR ≤ 2	51 (65.3%)	15 (51.7%)	35 (71.4%)	
TR ≥ 3	27 (34.7%)	14 (48.3%)	14 (28.6%)	
PASP (mmHg)	48.5 [38.8; 66.0]	46.0 [38.5; 58.5]	49.0 [40.0; 73.0]	0.190
TAPSE/PASP	0.40 ± 0.19	0.40 ± 0.15	0.40 ± 0.20	0.949

**Table 3 biomedicines-12-01710-t003:** Echocardiographic changes after TEER (median follow-up 6 months, IQR: 3 to 9 months). Significant *p*-values are in bold. LA, left atrial; LVEF, left ventricular ejection fraction; LVED, left ventricular end-diastolic volume; LVESV, left ventricular end-systolic volume; LVEDD, left ventricular end-diastolic diameter; LVESD, left ventricular end-systolic diameter; LV GLS, left ventricular global longitudinal strain; GWI, global work index; GCW, global constructive work; GWW, global wasted work; GWE, global work efficiency; LA, left atrial; TAPSE, tricuspid annular plane systolic excursion; RV FWS, right ventricular free wall longitudinal strain; PASP, pulmonary artery systolic pressure. (*) The first 16 variables were expressed as the means and standard deviations (**) Cut-offs according to the 2022 ESC Guidelines for the diagnosis and treatment of pulmonary hypertension [19].

	All Patients One Month before TEER (*)	All Patients Six Months after TEER (*)	*p*-Value
LVEF (%)	47.37 ± 14.26	43.74 ± 15.38	**<0.001**
LVEDV (mL)	130.59 ± 56.11	123.00 ± 53.24	**0.049**
LVESV (mL)	72.60 ± 47.17	71.33 ± 44.69	0.584
LVEDD (mm)	54.81 ± 8.96	53.30 ± 9.30	**0.013**
LVESD (mm)	40.09 ± 11.34	39.83 ± 11.57	0.742
GLS (%)	−13.28 ± 3.82	−12.28 ± 3.97	0.085
GWI (mmHg%)	1292.52 ± 608.54	1113.89 ± 489.49	**0.001**
GCW (mmHg%)	1647.15 ± 686.32	1438.56 ± 559.95	**0.009**
GWW (mmHg%)	173.50 ± 126.75	202.84 ± 104.63	0.062
GWE (mmHg%)	87.48 ± 8.18	85.11 ± 7.73	**0.007**
LA volume (mL)	70.52 ± 34.10	70.04 ± 36.62	0.841
LA strain reservoir (%)	12.03 ± 6.38	10.00 ± 4.56	**0.002**
TAPSE (mm)	17.80 ± 3.90	17.97 ± 3.96	0.687
RV FWS (%)	−19.36 ± 5.84	−20.01 ± 5.66	0.477
PASP (mmHg)	53.64 ± 18.64	43.86 ± 14.92	**<0.001**
TAPSE/PASP	0.39 ± 0.19	0.45 ± 0.18	**0.021**
TAPSE/PASP < 0.32 (**)	30 (38.46%)	13 (16.66%)	**0.006**
PASP > 56 mmHg	33 (42.30%)	16 (20.51%)	**0.006**
PASP > 40 mmHg	57 (73.07%)	46 (58.97%)	0.052

**Table 4 biomedicines-12-01710-t004:** Predictors of primary endpoint: heart failure hospitalization and cardiovascular death. Significant *p*-values are in bold. MR, mitral regurgitation; LVEF, left ventricular ejection fraction; LV GLS, left ventricular global longitudinal strain; GWI, global work index; GCW, global constructive work; GWW, global wasted work; GWE, global work efficiency; LA, left atrial; TAPSE, tricuspid annular plane systolic excursion; PASP, pulmonary artery systolic pressure; MR, mitral regurgitation.

Univariable Analysis	*p*-Value	Hazard Ratio
**Baseline clinical variable**		
MR type	0.280	1.67 (0.66–4.22)
Age	0.747	1.01 (0.97–1.05)
Sex	0.752	1.14 (0.51–2.56)
Diabetes	**0.045**	**2.25 (1.02–4.98)**
Known atrial fibrillation	0.194	1.84 (0.73–4.64)
Ischemic heart disease	0.057	2.19 (0.98–4.90)
Prior heart failure hospitalization	**0.017**	**3.69 (1.26–10.83)**
Prior cardiac surgery	0.404	0.68 (0.27–1.70)
Chronic kidney disease	0.187	1.70 (0.77–3.73)
Chronic obstructive pulmonary disease	0.523	1.42 (0.48–4.17)
**Baseline TTE variables (one month before TEER)**		
LVEF (%)	0.307	0.99 (0.96–1.01)
LV GLS (%)	0.353	1.05 (0.94–1.18)
GWI (mmHg%)	0.530	1.00 (1.00–1.00)
GCW (mmHg%)	0.474	1.00 (1.00–1.00)
GWW (mmHg%)	0.467	1.00 (0.99–1.00)
GWE (mmHg%)	0.925	1.00 (0.96–1.05)
LA volume (mL/m^2^)	0.837	1.00 (0.98–1.02)
LA strain reservoir (%)	0.122	0.94 (0.87–1.02)
TAPSE (mm)	0.545	0.97 (0.87–1.08)
RV FWS (%)	0.853	0.99 (0.88–1.11)
PASP (mmHg)	0.438	1.01 (0.99–1.03)
TAPSE/PASP	0.259	0.25 (0.02–2.79)
**TTE variables six months after TEER (IQR: 3 to 9 months)**		
LVEF (%)	0.377	0.99 (0.95–1.02)
LV GLS (%)	0.217	1.09 (0.95–1.24)
GWI (mmHg%)	0.230	1.00 (1.00–1.00)
GCW (mmHg%)	0.101	1.00 (1.00–1.00)
GWW (mmHg%)	0.267	1.00 (0.99–1.00)
GWE (mmHg%)	0.932	1.00 (0.94–1.06)
LA volume (mL)	0.818	1.00 (0.98–1.02)
LA strain reservoir (%)	**0.024**	**0.86 (0.76–0.98)**
TAPSE (mm)	0.305	0.94 (0.84–1.06
RV FWS (%)	0.552	1.06 (0.88–1.26)
PASP (mmHg)	0.247	1.02 (0.99–1.06)
TAPSE/PASP	**0.038**	**0.02 (0.00–0.80)**
TAPSE/PASP <0.47	**0.039**	**4.76 (1.08–21.02)**
LA strain reservoir <9%	**0.047**	**2.77 (1.01–7.59)**
MR grade 3 or 4	0.832	1.11 (0.43–2.87)

## Data Availability

The data presented in this study are available upon request from the corresponding author. The data are not publicly available due to ethical reasons.

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
