# Peer review of "Mitral Transcatheter Edge-to-Edge Repair and Clinical Value of Novel Echocardiographic Biomarkers: A Hypothesis-Generating Study"

_biomedicines, 2024, doi:10.3390/biomedicines12081710_

Round 1

Reviewer 1 Report (Previous Reviewer 1)

Comments and Suggestions for Authors

--

Author Response

We cannot provide a response to the reviewer as there are no additional comments.

Reviewer 2 Report (New Reviewer)

Comments and Suggestions for Authors

Caravaca et al reported their work named "Mitral Transcatheter edge-to-edge repair and clinical value of novel echocardiographic biomarkers: a hypothesis-generating study" and concluded "Conclusions. Echocardiography post-TEER reflects impairment in ventricular performance due to preload reduction and right ventricular to pulmonary artery coupling improvement. Short-term echocardiography after TEER identifies high-risk patients who could benefit from a close clinical follow-up. The prognostic significance of LA strain and the TAPSE/PASP ratio should be validated in subsequent large-scale prospective studies.". I have the following comments:

- Language revision is essential eg "All patients had a baseline and 6 months (IQR: 3-9 months) after TEER echocardiography" should be "All patients had a baseline and follow up echocardiography after TEER at 6 months".

- Please don't mention results in your abstract methods eg median, IQR, and number of included patients.

- I see in the abstract that you mentioned "Composite endpoint. You can keep that but you should report each endpoint as well. You reported "Survival probability" in the Y-axes of the Kaplan Meier curves, this should be revisited accordingly eg "Cumulative composite endpoint".

- Please spell out any abbreviation at their first time eg MR, and TAPSE/PASP in the abstract.

- Please add multivariable model and try to include TAPSE/PASP in it.

- Figure 4: Please specify the meaning of error bars eg 95% confidence interval or SD, etc. 

- Table S1: Please add number of included patients at the columns headings.

Comments on the Quality of English Language

Minor edits are needed as mentioned earlier.

Author Response

-Comment 1. Language revision is essential eg "All patients had a baseline and 6 months (IQR: 3-9 months) after TEER echocardiography" should be "All patients had a baseline and follow up echocardiography after TEER at 6 months".

*Response 1. We agree with this comment. We have revised this sentence according to the reviewer's suggestion.  See page 1, Abstract, lines 3-5: “All patients had echocardiography at baseline and again six months after TEER. They were monitored for a primary composite endpoint, consisting of heart failure hospitalization and cardiovascular death, over 13 months”.

-Comment 2. Please don't mention results in your abstract methods eg median, IQR, and number of included patients.

*Response 2. According to the reviewers' suggestions, we have not included methods (median, IQR) in the Abstract. However, we have emphasized the total number of patients included. See page 1, Abstract, lines 2-5.

-Comment 3. I see in the abstract that you mentioned "Composite endpoint. You can keep that but you should report each endpoint as well. You reported "Survival probability" in the Y-axes of the Kaplan Meier curves, this should be revisited accordingly eg "Cumulative composite endpoint".

*Response 3. We have decided to use a composite endpoint due to the study's small sample size and the limited number of events observed during the short clinical follow-up period. Additionally, in Figure 6, we have emphasized that survival probability is related to the composite primary endpoint (heart failure hospitalization and cardiovascular death). See page 13, Figure 6.

-Comment 4. Please spell out any abbreviation at their first time eg MR, and TAPSE/PASP in the abstract.

*Response 4. Thank you for pointing this out. We have reviewed and improved this aspect by spelling out all abbreviations at their first mention.

-Comment 5. Please add a multivariable model and try to include TAPSE/PASP in it.

*Response 5. Due to the limitations of the sample size, the cohort was analyzed as a single group, despite including various etiologies of mitral regurgitation. The low absolute number of events prevented us from performing a complex multivariate analysis. However, we included the etiology of mitral regurgitation in the multivariate analysis to highlight that "reservoir left atrial strain" and the "TAPSE/PASP ratio" remained significant prognostic echocardiographic variables regardless of the etiology of mitral regurgitation. See page 11, paragraph 5, lines 18-21.

- Comment 6. Figure 4: Please specify the meaning of error bars eg 95% confidence interval or SD, etc. 

*Response 6. Thank you for pointing this out. We have clarified this in the footer of Figure 4 (page 10): “Error bars on graphs represent the standard deviation of a dataset relative to the mean”.

- Comment 7. Table S1: Please add a number of included patients at the column headings.

*Response 7.  We have added the number of patients to the column headings. Please refer to Table S1 in the Supplementary Materials.

Reviewer 3 Report (New Reviewer)

Comments and Suggestions for Authors

See attached file

Author Response

-Comment 1. Novelty and importance. The clinical significance of the study is limited as it concerns a retrospective study on a limited number of patients (n = 78) and events with limited FU duration (median 13 m). Thus indeed the results are at its best hypothesis-generating. Nonetheless there is an interesting aspect in the authors’ study design. The percentage of patients with organic MR exceeds the percentage in earlier studies in the field. The study depicts data in a mixed MR study group but with a preponderance of organic MR. It is an aspect of novelty.

*Response 1: We agree with this comment. First, this retrospective, single-center, real-life study comprises a limited and heterogeneous population with mitral regurgitation caused by different etiologies and a relatively short follow-up period. Because of the study's small sample size and the limited number of events observed during the short clinical follow-up period, our analysis serves as hypothesis-generating research. However, most previous retrospective and single-center studies analyzing changes in cardiac remodeling in patients treated with TEER include a similar number of patients (see Bibliography). Additionally, in contrast to previously published studies, and as a novel aspect, the present registry mainly includes patients with organic MR, whose clinical evolution after TEER remains unknown to date.

-Comment 2. Follow-up. Indeed the FU is limited. Table 4 nicely illustrates that the significant prognostic value of LA strain reservoir and TAPSE/PASP was restricted to the values obtained 6-m after the TEER procedure. The median FU between the month 6 echocardiography and the events was 8 months (Fig 1) illustrating the short-term observation period. The short-term prognosis is scientifically of interest but those data still do not guarantee similar long-term prognostic parameters.  Moreover although statistically significant (Table 4, p = 0.038) the ratio TAPSE/PASP as a continuous parameter cannot be promoted for follow-up and clinical management as the CIs are much too broad [HR 0.02 (0.00-0.80)]. Maybe the clinical parameters DM and prior HF hospitalization are at least as important for clinical management. Moreover it concerns results from univariate Cox models. Multivariable-adjusted Cox models may yield other determinants than LA strain reservoir or TAPSE/PASP or with completely other HRs for the incriminated parameters. Thus more than one word of caution is mandatory! See also comment 5.

*Response 2. We agree with this comment. This study comprises a limited and heterogeneous population with MR caused by different etiologies and a relatively short follow-up period. We have added this aspect in Limitations. See page 16, Limitations, paragraph 5, lines 36-37: “Our results may encourage further studies in larger cohorts with extended follow-up periods”. On the other hand, the low absolute number of events prevented us from performing a complex multivariate analysis. We included the etiology of mitral regurgitation in a small multivariate analysis to highlight that "reservoir left atrial strain" and the "TAPSE/PASP ratio" remained significant prognostic echocardiographic variables regardless of the etiology of mitral regurgitation (see page 11, paragraph 5, lines 18-21). Finally, due to the study's limitations, which serve as hypothesis-generating research, we have revised the Conclusions to avoid being overly definitive. See page 17, paragraph 2, lines 6-7: “We hypothesized the TAPSE/PASP ratio, and LA reservoir obtained in short-term TTE after TEER are important prognostic indicators that could be associated with a higher rate of adverse events”.

-Comment 3. There is a high drop-out: only 78/110 could be analyzed.

*Response 3: This is a limitation of the study. Thirty-two patients were excluded due to an incomplete echocardiographic evaluation or insufficient quality echocardiography imaging. Novel echocardiographic biomarkers such as myocardial work indices require an adequate acoustic window and a comprehensive echocardiographic assessment for accurate measurement.

-Comment 4. Methods. We are not dealing with an invasive study. Peak systolic LV pressure was taken as the brachial cuff SBP. From physiology point-of-view not correct but anyhow in elderly patients (as in this study) it is known that the amplification of SBP to the peripheral arteries is blunted. Therefore it might not be a bad choice.

*Response 4: We opted to measure non-invasive left ventricular pressure using brachial artery SBP because this measurement is routinely performed for all echocardiograms conducted in our imaging department.

-Comment 5: The Tables learn that the MR patients are complicated by a long list of co-morbidities. It is another reason why univariate Cox models focusing on echocardiography parameters only may dramatically fail in such patients (multivariable approach is mandatory in future studies).

*Response 5: The low absolute number of events prevented us from performing a complex multivariate analysis. We included the etiology of mitral regurgitation in a small multivariate analysis to highlight that "reservoir left atrial strain" and the "TAPSE/PASP ratio" remained significant prognostic echocardiographic variables regardless of the etiology of mitral regurgitation (see page 11, paragraph 5, lines 18-21). However, in complete agreement with the reviewer, a multivariable approach is essential for future studies involving larger cohorts and a greater number of events.

-Comment 6: Table 1. The reviewer is a little bit surprised to read that clinical baseline parameters did not differ between functional and organic MR. Was it also within the authors’ expectations?

*Response 6: We cannot be certain why the clinical baseline parameters did not differ between functional and organic MR. It is possible that the functional cohort was smaller and not representative. Additionally, the functional cohort differs from the COAPT trial population due to the significant atrial MR mechanisms present in our study.

-Comment 7: Fig 4. It is clear that despite highly significant (p < 0.01) differences for GCW and GWE between pre- and post-TEER the clinical differences remain limited.  Did the authors expect more from the prognostic value of the global work indices?

*Response 7: Thank you for pointing this out. Previous studies have highlighted the potential of these echocardiographic biomarkers, and future research should consider their prognostic value in this clinical scenario. However, it appears that a decline in ventricular function (such as LVEF, strain, and myocardial work) is primarily associated with the reduction in mitral regurgitation and may not necessarily indicate a poor prognosis. Additionally, some authors have reported that GLS and myocardial work indices may recover after TEER during extended follow-up.

This manuscript is a resubmission of an earlier submission. The following is a list of the peer review reports and author responses from that submission.

Round 1

Reviewer 1 Report

Comments and Suggestions for Authors

This retrospective cohort study based on a very limited number of patients (n=78) indented to determine prognostic indicators of high risk patients. The manuscripts has some flaws which have to be corrected:

- besides the low number of patients additional factors may play an important role for the identification of prognostic indicators: i.e. success rate (65.4%) and complication rate (10.2%) which are are not included in the regression analyses but may influence the primary endpoint (mortality resp. hospitalisation);

- the conclusion and the abstract do not reflect the limited  power of the results.

- The statement "patients with functional MR exhibited a non-significantly higher LA volume ..." may lead to not relevant conclusions.

Minor: fig. 4: media instead of median

Reviewer 2 Report

Comments and Suggestions for Authors

Dear Authors,

I wish to congratulate you with the presented manuscript. The submitted analysis raise my significant concerns. Please find the enclosed suggestions:

1.     “there was an improvement in the pulmonary artery systolic pressure and tricuspid annular plane systolic excursion/pulmonary artery systolic pressure (TAPSE/PASP) ratio” 

what does the improvement mean? The presented parameters may increase or decrease, clinical improvement may be observed 

2.     The one of the major concerns are related to inaccuracy of the great value for conclusion drawing:

a.     In abstract the information appear: “All patients had a baseline and short-term echocardiography after TEER and were followed up for the primary composite endpoint (heart failure hospitalization and cardiovascular death) over a median period of 13 months”

b.     In material section: “For this analysis, only patients with complete transthoracic echocardiography (TTE) at baseline (one month before TEER) and short-term follow-up (median: 6 months, interquartile range [IQR]: 3 to 9 months) were included”. Followed by: “Demographic and clinical information were collected from the patient’s electronic medical records. Patients were monitored for the primary composite endpoint, defined as heart failure hospitalization and cardiovascular death. The median follow-up period was 13 months after TEER (IQR: 5 to 20 months)

3. Table 4 raise my concerns and I would advise to share the data base to the statistician related to the journal or if possible to the reviewers:

a.     LVEDD (mm) 54.81 ± 8.96 vs 53.30 ± 9.30 p=0.013 (rise my concerns)

b.     GWE (mmHg%) 87.48 ± 8.18 vs 85.11 ± 7.73 p=0.007 (rise my concerns)

4. Multivariable analysis:

a.     First of all: the echocardiographic results are collected within 6 months after procedure and the  models are created for 12 months results prediction which is highly misleading (!)

b.     The results of multivariable analysis are not presented clearly

c.     The multivariable models are presented regarding to etiology but not to the whole group

d.     I find it unclear as the univariable model was created for the whole group

5. The alarming results regarding end-points were found in relatively high numbers as follow “The primary composite endpoint occurred in 27 patients (34.6%).” That is in contrary to first sentences in discussion as follow: “TEER is a safe and effective treatment in symptomatic patients with MR.”

kind regards